# Determinants Affecting the Awareness of Hypertension Complications within the General Population in Saudi Arabia

**DOI:** 10.3390/healthcare12161674

**Published:** 2024-08-22

**Authors:** Muffarah Hamid Alharthi, Elhadi Miskeen, Eman Abdullah Alotaibi, Ibrahim Awad Eljack Ibrahim, Mohannad Mohammad S. Alamri, Mohammad S. Alshahrani, Dina S. Almunif, Abdullah Almulhim

**Affiliations:** 1Department of Family and Community Medicine, College of Medicine, University of Bisha, Bisha 61922, Saudi Arabia; ieljack@ub.edu.sa (I.A.E.I.); malamri@ub.edu.sa (M.M.S.A.); 2Department of Obstetrics and Gynecology, College of Medicine, University of Bisha, Bisha 61922, Saudi Arabia; emiskeen@ub.edu.sa; 3Department of Family and Community Medicine, College of Medicine, Qassim University, Qassim 51452, Saudi Arabia; e.alotaibi@qu.edu.sa; 4Endocrine and Diabetes Centre, Armed Forces Hospital South Region, Khamis Mushait 62413, Saudi Arabia; malshahrani124@hotmail.com; 5Department of Family and Community Medicine, College of Medicine, King Saud University, Riyadh 12372, Saudi Arabia; d.almunif@gmail.com; 6Department of Family and Community Medicine, College of Medicine, King Faisal University, Alahsa 31982, Saudi Arabia; abdullah05039@hotmail.com

**Keywords:** hypertension complications, awareness, Bisha, general population, Saudi Arabia

## Abstract

Background: Hypertension imposes a significant public health burden. An increased awareness of hypertension complications within a population can positively impact patient care and prevent complications. This study seeks to assess the awareness of hypertension complications among the population of Bisha in Saudi Arabia in 2020. Methods: A cross-sectional study was conducted in 2020. A validated self-administered online-based questionnaire was sent to a sample of the adult population of Bisha to measure their awareness of hypertension complications. Results: Almost three-quarters of the population (72.2%) were aware of hypertension complications. The awareness level was significantly higher among male participants (*p* < 0.001), those aged 31–40 years, those who were married, those working as police officers or in civilian jobs, those living in urban areas (*p* = 0.04), those with a university-level education (*p* = 0.03), those with a medium family income (SAR 5000–14,999) (*p* = 0.001), and those with a history of hospitalization because of causes other than hypertension (*p* = 0.05). Marital status was independently predictive of awareness (B = 0.851, Wald test = 12.179, *p* = 0.000) among the respondents. Conclusion: The study concludes that the awareness of hypertension complications among the Bisha population in Saudi Arabia was deemed acceptable. Factors such as marital status, age, gender, a family history of hypertension, the duration of hypertension, and medication adherence positively influenced this awareness and served as predictors of hypertensive awareness. The findings highlight the importance of health authorities in ensuring the widespread awareness of hypertension complications, particularly among hypertensive individuals.

## 1. Introduction

Hypertension is a serious public health condition that can increase the risk of disease in different organs. It is a known cause of premature death all over the world, with one-quarter of men and one-fifth of women—more than a billion people—having this disease [1]. This is one of the most important public health problems in the world. Cardiovascular diseases resulting from atherosclerosis are a main cause of death, and hypertension is an important modifiable risk factor for the development of atherosclerosis [2].

The burden of hypertension is felt disproportionately in developing countries, where most of the hypertensive cases are found, because of widely distributed disease determinants in these countries [1]. Hypertension is diagnosed when systolic blood pressure readings measured on two different days are ≥140 mmHg and/or the diastolic blood pressure readings on both days are ≥90 mmHg [3,4]. The prevalence of hypertension differs in different parts of the world, with the least prevalence in rural India (3.4% of men and 6.8% of ladies) and the most elevated prevalence in Poland (68.9% of men and 72.5% of women). The awareness of hypertension was registered in 46% of studies and fluctuated from 25.2% in Korea to 75% in Barbados [5].

The prevalence of hypertension among adults was higher in low- and middle-income countries (31.5%, 1.04 billion people) than in high-income countries (28.5%, 349 million people). Differences in hypertension determinants, for example, high sodium consumption, low potassium consumption, obesity, alcohol intake, low levels of exercise, and unhealthy foods, may clarify a portion of the regional variations in the hypertension prevalence rate. Notwithstanding its expanding prevalence, the extent of hypertension mindfulness, management, and blood pressure control is low, especially in low- and middle-income nations, and there are inadequate inclusive measurements of the monetary effects of hypertension [6].

More than one-fourth of the world’s adult population (26.4%) has hypertension (95% CI 26.0–26.8%), and its prevalence is expected to increase to 29% by 2025, especially in developing countries where the number of hypertensive adults will increase by 60% [7]. This high prevalence of hypertension contributes to the public health burden all over the world, leading to heart disease, stroke, death, and disability-adjusted life years lost worldwide [8].

An awareness of hypertension is of great importance. A study conducted in Matabeleland in Zimbabwe showed that the knowledge of hypertension was poor, with 64.8% of respondents stating that stress was its main cause, 85.9% stating that palpitations were a symptom of hypertension, and 59.8% stating that adding salt to meals is another cause. The more education respondents had received, the more likely they were to be knowledgeable about hypertension (odds ratio for secondary education, 3.68 [95% CI: 1.61–8.41], and for tertiary education, 7.52 [95% CI: 2.76–20.46]), compared to those without a formal education [9]. Moreover, in a study conducted in Yazd, Iran, 49.7% of individuals with hypertension knew about their sickness. In the adjusted model, the elderly, females, and those with diabetes mellitus were positively associated with a higher understanding. Having healthcare coverage corresponded with an awareness of hypertension [10].

A hospital-based cross-sectional study, targeting the Riyadh population of Saudi Arabia, showed that of 384 participants, nearly half had a good level of awareness of hypertension complications (46.61%), while 37.50% had moderate knowledge [11].

Previous studies conducted in Saudi Arabia investigated the knowledge, attitudes, and practices regarding hypertension in general, or concentrated on the control and treatment of hypertension [12,13,14]. However, limited data are available on the Saudi population’s awareness of hypertension complications. Awareness studies to determine the knowledge of the Saudi population are of great importance, especially those targeting the adult population among whom hypertension is prevalent. The population’s awareness of hypertension complications will have tremendous effects by helping with patient care and avoiding complications. Because of this, more studies are required to provide evidence-based information to health professionals, which will enable them to design appropriate strategies to improve the awareness of hypertension complications in the area of the study. The goal of this study is to determine the awareness of hypertension complications among the Bisha population in Saudi Arabia.

## 2. Materials and Methods

### 2.1. Ethical Statement

Ethical clearance was obtained from the University of Bisha College of Medicine-Research Ethics Local Committee (UBCOM/H-06-BH-027). Informed consent was obtained from all participants. The significance and points of the investigation were disclosed and consent to partake in the study was obtained from the participants.

### 2.2. Study Design

A community-based cross-sectional awareness study was conducted from August 2020 to October 2020 among the Bisha population in Saudi Arabia. A validated online questionnaire was sent to the participants using Google Forms to collect the information. The questionnaire originated based on the possible awareness of hypertension complications and was pretested on 50 subjects like the study members for validation purposes. The questionnaire included questions ascertaining personal characteristics and awareness of the complications of hypertension.

### 2.3. Sample Size

The sample size was calculated using the formula Z^2^ p (1 − p)/d2 with the assumption that *p* = 50% and that there was a 95% level of confidence and a 5% margin of error (d). The sample included 384 individuals. We increased the sample size to 461 to enhance our study’s precision.

### 2.4. Study Site

Bisha is in south-west Saudi Arabia. It is located in the Asir region, and it is the capital of the province of Bisha. It stands at an altitude of about 2000 feet above sea level. Bisha University, a public university that was founded many years ago, is also located in the city. Bisha features around 240 villages [15].

### 2.5. Study Population

The study population consisted of individuals living in Bisha aged ≥18. Those with a health professional background were excluded from this study. The study utilized a community-based cross-sectional design to assess the awareness of hypertension complications among the population of Bisha in Saudi Arabia. The sample selection process aimed to capture a representative sample of the adult population of Bisha while ensuring ethical standards and data accuracy. The study utilized a convenience sampling technique to select participants.

### 2.6. Study Instrument

A self-administered questionnaire was created in Arabic based on the most recent available information after a deep search of the medical literature. A pretest of the questionnaire was conducted on 50 people, not included in the study sample, to test its validity and to check for any needed modifications before finalizing the questionnaire. The questionnaire was divided into three parts: the first part ascertained the sociodemographic characteristics of the participants, the second determined the respondents’ source of information about hypertension complications, and the third measured the participants’ level of awareness.

To measure their level of awareness, respondents were able to answer 13 questions with “yes,” “no,” or “I don’t know.” Correct answers were given 1 point and incorrect answers or “I don’t know” answers were given 0 points. The total awareness score ranged from 0 to 13. These scores were categorized into good awareness, aware, and not aware, based on 50% and 75% cut-off points of the total possible score.

### 2.7. Statistical Analysis

Data on the awareness of hypertension complications were analyzed using SPSS 23. Descriptive and inferential statistical analyses were performed, and the association between sociodemographic characteristics and the awareness of hypertension complications was tested. Numbers and percentages were used to present nominal data. A univariate analysis was run to study the independent associations of variables (age, gender, residence, marital status, level of education, job, income, source of information, history of hypertension and hospitalization, and family history of hypertension) with the awareness level of hypertension complications using the chi-squared test. A 95% confidence interval (CI) was performed to measure association strength and a *p*-value of less than 0.05 was considered statistically significant. Logistic regression analysis was carried out to identify determinants of the awareness level of hypertension complications. Independent variables included in the model were participants’ age, gender, residence, marital status, level of education, job, income, history of hypertension, and family history of hypertension. A dependent variable introduced to the model was the awareness level (i.e., “aware” vs. “not aware”). Determinants with a screening significance of *p* < 0.05 in the univariate analyses were selected for multivariate analyses (binary logistic regression). All *p*-values were two-tailed and were considered statistically significant at <0.05.

The scores for awareness of hypertension complications were then summed up to generate an overall score for each participant. Awareness levels were then categorized depending on the total score into “not aware” for respondents scoring <50%, “aware” for those scoring ≥50 and ≤74%, and “good awareness” for those scoring ≥75%.

## 3. Results

### 3.1. Personal Characteristics

A total of 461 questionnaires from the study population were included in this study, with a response rate of 89%. Of those, 299 (64.9%) were males and the rest (162, 35.1%) were females (Table 1). Most of the respondents (34.5%) were aged between 31 and 40 years. The distribution of the remaining personal characteristics is shown in Table 1.

### 3.2. Family History and Medical History of Hypertension

As shown in Table 2, more than half of the respondents had a family history of hypertension (50.3%) and 30.2% of them had a hypertensive parent. Almost half of the respondents had been diagnosed with hypertension (41.6%). In total, 16.1% of them had been diagnosed 1–5 years ago and 30.1% had hypertension complications. Of those who had complications, only 21.9% of them mentioned that their primary healthcare doctor had discussed hypertension complications with them and 13.1% of them developed complications after diagnosis. The most common complication mentioned by the respondents was chronic headache (4.3%). A total of 8.5% of the respondents had been hospitalized during the last year—2.6% of them because of hypertension and 1.7% of them because of a stroke complication.

### 3.3. Respondents’ Level of Awareness of Hypertension Complications

#### 3.3.1. Source of Information

The respondents’ most common source of information about hypertension complications was a collection of “healthcare worker, social media, and Internet” sources (16.9%), followed by healthcare workers (12.4%).

#### 3.3.2. Awareness of Hypertension Complications

The overall mean awareness score was 8.1 ± 2.63. Almost three-quarters of the population (72.2%) were aware of hypertension complications (52.2% were aware and 20% had good awareness), while the remaining 27.8% were unaware (Figure 1). Unawareness was more apparent in response to questions asking whether hypertension can lead to complications such as dementia (where only 27.5% of the respondents answered “yes” and 54.0% of them responded “I don’t know”), impotence (where 13.4% of the respondents answered “no” and 49.2% of them responded “I don’t know”), and diabetes mellitus (where 13.7% of the respondents answered “no” and 47.3% of them responded “I don’t know”).

### 3.4. Associations between the Respondents’ Characteristics and Their Awareness of Hypertension Complications

Significant associations were found between some personal characteristics and awareness scores. Awareness scores were significantly higher among male participants (*p* < 0.001), those aged 31–40 years, those who were married, those working as police officers or in civilian jobs, those living in urban areas (*p* = 0.04), those with a university-level education (bachelor’s degree) (*p* = 0.03), those with a family income of SAR 5000–14,999 (*p* = 0.001), and those with a history of hospitalization because of causes other than hypertension (*p* = 0.05). No significant associations were present between the awareness scores and a family history of hypertension (*p* = 0.71), the duration of hypertension (*p* = 0.42), the presence of hypertension complications (*p* = 0.37), information sources (*p* = 0.15), the primary healthcare doctor discussing hypertension complications with the patient (*p* = 0.19), and the occurrence of hypertension complications after diagnosis.

Personal characteristic variables assessed for associations with the outcome variables during the univariate analysis were re-entered into a final multivariate model using binary logistic regression analysis. In the multivariate analysis, marital status was independently predictive of awareness (B = 0.851, Wald test = 12.179, *p* = 0.000) among the respondents.

### 3.5. Regression Analysis

Regression analysis interpretations (Table 3) provide insights into the factors that influence the awareness of hypertension complications among the patients in the study.

Patients who are one year older have 2.27 times higher odds of being aware of hypertension complications compared to younger patients. This relationship is statistically significant (*p* < 0.001, CI 1.45–3.56, OR 2.27), suggesting that age is an important factor in predicting awareness.

Male patients have 1.60 times higher odds of being aware of hypertension complications compared to female patients. This association is statistically significant (*p* = 0.027, CI 1.07–2.39, OR 1.60), indicating that gender plays a role in awareness.

The education level of the patients does not have a statistically significant relationship with their awareness of hypertension complications (*p* = 0.407, CI 0.82–1.65, OR 1.16). Therefore, there is insufficient evidence to suggest that education level affects awareness.

Patients with a family history of hypertension have 1.88 times higher odds of being aware of hypertension complications compared to those without a family history. This relationship is statistically significant (*p* = 0.013, CI 1.12–3.15, OR 1.88), suggesting that a family history of hypertension influences awareness.

The income level of the patients does not have a statistically significant relationship with their awareness of hypertension complications (*p* = 0.102, CI 0.52–1.11, OR 0.76). Thus, income level is not a significant predictor of awareness in this study.

For every additional year a patient has had hypertension, their odds of being aware of hypertension complications decrease by a factor of 0.40. This negative relationship is statistically significant (*p* < 0.001, CI 0.25–0.63, OR 0.40), indicating that a longer duration of hypertension is associated with lower awareness.

Patients who are more adherent to their medication have 2.76 times higher odds of being aware of hypertension complications compared to those with lower adherence. This relationship is statistically significant (*p* < 0.001, CI 1.62–4.71, OR 2.76), indicating that medication adherence is an important predictor of awareness.

## 4. Discussion

This study aimed to assess the awareness of hypertension complications among the Bisha population in Saudi Arabia. Despite some limitations in self-reported questionnaire responses, this study provides valuable insights for healthcare decision-makers and researchers addressing hypertension-related public health challenges.

The overall mean score of the awareness of hypertension complications in this study was moderate; almost three-quarters of the participants had awareness (72.2%) and only 27.8% had poor awareness. This finding was consistent with a 2017 study in Tanzania, which found that more than 50% of participants were aware that hypertension can cause complications such as heart disease and stroke, while most of the participants were aware that compliance with regular treatment for hypertension can prevent or slow down the occurrence of complications [16]. A study of hypertensive patients and normotensive patients in Pakistan showed that both groups knew that hypertension could prompt stroke and cardiac illnesses, while the greater part of them was not aware of other significant complications of hypertension, for example, kidney diseases, atherosclerosis, and eye problems.

A study in Riyadh, Saudi Arabia, in 2017 showed that among 384 participants, nearly half of them had a good level of awareness regarding hypertension complications (46.61%) and 37.50% had moderate knowledge [17]. A study of patients with suspected hypertension visiting the outpatient department at Rohilkhand Hospital in India found that most patients with hypertension knew that they were more susceptible to cardiac diseases (66.7%), renal disorders (35.71%), brain damage (34.7%), and others (19%) [18]. Another study in 2014 investigated the impact of patient knowledge of hypertension complications on compliance with hypertension medications in Algeria, showing that 73.7% of the participants were knowledgeable about hypertension complications [19].

Moreover, the findings of this study reveal that the study population was more enthusiastic about gaining information about hypertension complications from healthcare workers, social media, and Internet sources. Our finding differs from that of Akter et al.’s study [20], which showed that the major wellsprings of knowledge are TV and the web (82.6%), books/magazines and journals (59%), knowledge-based sources, for example, medical experts/talks and seminars (72%), and colleagues and relatives (81.4%).

This regression analysis provides insights into the factors associated with the awareness of hypertension complications among the patients in this hypothetical study. It highlights the importance of age, gender, family history, hypertension duration, and medication adherence as potential influencers of awareness. These findings are consistent with studies stating that patients of increased age [21], males [22], patients with a family history of hypertension [23], patients with a greater duration of hypertension [24,25], and patients who are more adherent to their medication [26,27,28,29] have higher odds of being aware of hypertension complications. These insights can guide targeted health education interventions to improve the awareness and management of hypertension in the community.

Expanding knowledge about hypertension in Saudi society can be achieved through public awareness campaigns, community workshops, school-based education programs, workplace wellness initiatives, health screenings, the engagement of religious leaders, digital health platforms, peer education programs, partnerships with healthcare providers, and continuous evaluation and feedback mechanisms. These efforts aim to raise awareness, promote healthy behaviors, and empower individuals to prevent and manage hypertension effectively.

There were several limitations to this study. First, self-reported questionnaire responses may not be entirely accurate and should be viewed with caution as they may be subjective. This may limit the reliability of the results because of the possible tendency of the participants to paint a more positive picture than would be reflected by other data collection tools. Some respondents may have answered in such a way as to please the researcher or in a manner that they perceived as correct (social desirability bias), thus not reflecting their real response. However, we are hopeful that the anonymity of the survey motivated the participants to be honest in their replies.

With implications of potential deviations from perfect representativeness on the generalizability of the research findings, it is important to address that a representative sample is a limitation. This discussion is crucial for contextualizing the results, guiding the application of the research, and planning future studies to address any gaps in knowledge.

We believe that this study addresses a major public health problem that challenges healthcare providers and the entire population of Saudi Arabia. Despite its limitations, we believe that this study could be a reasonable source of information for researchers and healthcare decision-makers.

## 5. Conclusions

The findings of the study provide valuable insights into the awareness levels of hypertension complications among the population of Bisha in Saudi Arabia and suggest strategies for enhancing awareness and improving health outcomes related to hypertension. With hypertension being a significant public health concern globally, understanding the level of awareness within specific communities is vital for effective prevention and management strategies. By improving awareness of hypertension complications, healthcare stakeholders can contribute to better health outcomes, reduced disease burden, and enhanced quality of life for individuals living with hypertension in Saudi Arabia.

## Figures and Tables

**Figure 1 healthcare-12-01674-f001:**
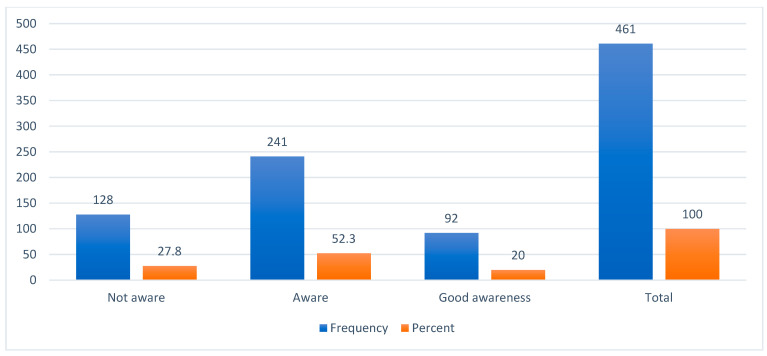
The frequency and percentage distribution of the respondents by their total score for awareness of hypertension complications.

**Table 1 healthcare-12-01674-t001:** The sociodemographic structure distribution of the respondents.

Age Group	Frequency	Percent
18–30	141	30.6
31–40	159	34.5
41–50	86	18.7
51–60	59	12.8
>60	16	3.5
Total	461	100.0
**Gender**	**Frequency**	**Percent**
Male	299	64.9
Female	162	35.1
Total	461	100.0
**Residency**	**Frequency**	**Percent**
Urban	350	75.9
Rural	111	24.1
Total	461	100.0
**Level of education**	**Frequency**	**Percent**
Illiterate	4	0.9
Primary school	10	2.2
Intermediate school	13	2.8
Secondary school	97	21.0
Diploma	50	10.8
Bachelor’s degree	249	54.0
Postgraduate	38	8.2
Total	461	100.0
**Marital status**	**Frequency**	**Percent**
Single	89	19.3
Married	349	75.7
Divorced	11	2.4
Widowed	10	2.2
Separated	2	0.4
Total	461	100.0
**Occupation**	**Frequency**	**Percent**
Police officer	134	29.1
Civilian job	191	41.4
Private sector	35	7.6
Business person	12	2.6
Unemployed	89	19.3
Total	461	100.0
**Family income**	**Frequency**	**Percent**
<3000	38	8.2
3000–4999	37	8.0
5000–9999	138	29.9
10,000–14,999	140	30.4
15,000–24,999	80	17.4
≥25,000	28	6.1
Total	461	100.0

**Table 2 healthcare-12-01674-t002:** The frequency and percentage distribution of the respondents by family history and medical history related to hypertension.

Family History of Hypertension	Frequency	Percent
Yes	232	50.3
No	229	49.7
Total	461	100.0
**Family member with hypertension**	**Frequency**	**Percent**
Father	80	17.4
Mother	59	12.8
Brother or sister	12	2.6
Grandparent	9	2.0
Child	3	0.7
Spouse	11	2.4
Uncle or aunt	8	1.7
Total	182	39.5
**If you have hypertension, when were you diagnosed?**	**Frequency**	**Percent**
<1 year	47	10.2
1–5 years	74	16.1
6–10 years	30	6.5
>10 years	41	8.9
Total	192	41.6
**Do you have complications of hypertension?**	**Frequency**	**Percent**
Yes	142	30.8
No	7	1.5
I don’t know	43	9.3
Total	192	41.6
**Did a primary healthcare doctor discuss the complications of hypertension with you?**	**Frequency**	**Percent**
Yes	101	21.9
No	86	18.7
Total	187	40.6
Not answered	274	59.4
Total	461	100.0
**Did you get any complications of hypertension after your diagnosis?**	**Frequency**	**Percent**
Yes	61	13.2
No	81	17.6
Total	142	30.8
**Which complications of hypertension did you get after your diagnosis?**	**Frequency**	**Percent**
Chronic headache	20	4.3
Neck pain	4	0.9
Increased blood sugar	3	0.7
Stroke	6	1.3
CVS disease	1	0.2
Blindness	4	0.9
Heart failure	2	0.4
Dizziness	1	0.2
Renal failure	1	0.2
Nervousness	3	0.7
Premature delivery	1	0.2
Pre-eclampsia	1	0.2
Total	47	10.2
**Have you been hospitalized in the last year?**	**Frequency**	**Percent**
Yes	39	8.5
No	422	91.5
Total	461	100.0
**What were the causes of your hospitalization?**	**Frequency**	**Percent**
Primary hypertension	12	2.6
Stroke	8	1.7
Headache	3	0.7
Hypotension	1	0.2
Pre-eclampsia	1	0.2
Renal failure	3	0.7
Abortion	1	0.2
Heart failure	3	0.7
Renal stone	2	0.4
IBS	1	0.2
Hemorrhoid	2	0.4
Hyperglycemia	2	0.4
Total	39	8.5
**Was your hypertension controlled in the hospital?**	**Frequency**	**Percent**
Yes	31	6.7
No	1	0.2
I don’t know	7	1.5
Total	39	8.5
**Was there any relation between your hospitalization and hypertension?**	**Frequency**	**Percent**
Yes	32	6.9
No	52	11.3
I don’t know	28	6.1
Total	112	24.3
Total	461	100.0

**Table 3 healthcare-12-01674-t003:** Regression analysis for factors influencing awareness of hypertension complications among the participants (*n* = 461).

Factor	95% CI	Odds Ratio	*p*-Value
Age (Years)	1.45–3.56	2.27	0.001 *
Gender (Male)	1.07–2.39	1.60	0.027 **
Education Level	0.82–1.65	1.16	0.407 **
Family History	1.12–3.15	1.88	0.013 **
Income Level	0.52–1.11	0.76	0.102 *
Duration (Years)	0.25–0.63	0.40	0.001 *
Medication Adherence	1.62–4.71	2.76	0.001 **

Legend: * logistic regression. ** Linear regression.

## Data Availability

The datasets during and/or analyzed during the current study are available from the corresponding author upon reasonable request.

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
