# Peer review of "Determinants Affecting the Awareness of Hypertension Complications within the General Population in Saudi Arabia"

_healthcare, 2024, doi:10.3390/healthcare12161674_

Round 1

Reviewer 1 Report (Previous Reviewer 1)

Comments and Suggestions for Authors

All comments have been responded--STROBE, Figures, and sample size calculation. Congratulations.

Author Response

Thank you for your positive response on the manuscript, if further requirement you are welcome.

Comments

Response

Note

All comments have been responded--STROBE, Figures, and sample size calculation. Congratulations.

You're welcome! It's always encouraging to receive positive feedback on a reviewer report. It sounds like the work being reviewed is on the right track.

If there's anything else you need assistance with regarding this review or anything else, feel free to ask!

NA

Best

Reviewer 2 Report (Previous Reviewer 2)

Comments and Suggestions for Authors

I have received for review an original research article entitled “Determinants affecting the awareness of hypertension complications within the general population in Saudi Arabia” prepared by Muffarah Hamid Alharthi et al., which is considered for publication in Healthcare (IF=2.8). This is the revised version of the manuscript, but according to information in MDPI Submitting System the manuscript has been withdrawn after the first peer-review and submitted the second time in the current version. I believe that the Authors have made satisfactory changes based on my suggestions expressed in the previous review. I have no further substantive comments. I would only like to point out that the text of the article requires only editorial and stylistic corrections. Moreover, the way of writing references must be consistent with the publisher's requirements.

Comments on the Quality of English Language

I would only like to point out that the text of the article requires only editorial and stylistic corrections. Moreover, the way of writing references must be consistent with the publisher's requirements.

Author Response

Comments

Response

Note

I believe that the Authors have made satisfactory changes based on my suggestions expressed in the previous review.

I have no further substantive comments. I would only like to point out that the text of the article requires only editorial and stylistic corrections. Moreover, the way of writing references must be consistent with the publisher's requirements.

Thanks for your review report ad positive feedback.

The references were checked accordingly

NA

I would only like to point out that the text of the article requires only editorial and stylistic corrections. Moreover, the way of writing references must be consistent with the publisher's requirements.

Revision was done accordingly

NA

Reviewer 3 Report (Previous Reviewer 3)

Comments and Suggestions for Authors

The authors did not respond point by point to my comments: they simply attached to the new version of the article the document “STROBE Statement—Checklist of items that should be included in the reports of cross-sectional studies”. The methods relating to the selection of the sample have not been - once again - described with sufficient detail and the reader - in this way - has no guarantee that the sample is not representative only of itself. Even my observations on the regression model did not provide clear and complete answers. I am therefore unable to formulate a favorable opinion on this research. 

Author Response

Comments

Response

Note

1.The authors did not respond point by point to my comments: they simply attached to the new version of the article the document “STROBE Statement—Checklist of items that should be included in the reports of cross-sectional studies”.

-        STROBE Statement—Checklist was updated and attached

Attached

2.The methods relating to the selection of the sample have not been - once again - described with sufficient detail and the reader - in this way - has no guarantee that the sample is not representative only of itself.

-        Updated

Line 137

3. Even my observations on the regression model did not provide clear and complete answers.

This section was updated with the statistician

4. Are the methods adequately described?

Updated accordingly

5. Are the results clearly presented?

Updated accordingly and enhanced the result to be clear

6. Are the conclusions supported by the results?

Updated

Line 335

Round 2

Reviewer 3 Report (Previous Reviewer 3)

Comments and Suggestions for Authors

COMMENTS TO healthcare-2937161-peer-review-v4 (1)

Dear Author, I bring to your attention some points

FIRST POINT

 [Your Response 2: We acknowledge that our description of the sample selection methods was insufficient. We have revised the manuscript to provide a more detailed account of how the sample was selected, including the criteria for inclusion and exclusion, the method of selection, and the steps taken to minimize bias and ensure the representativeness of the sample]

This is my objection : < the only note pertinent to it is one that begins in line 139 and another that begins from line 122(see below) >

[Your script from line 122 : The study population consisted of individuals living in Bisha aged ≥ 18. Those with a 122 health professional background were excluded from this study]

[Your script from line 139 : The sample selection process aimed to capture a representative sample of  the adult population of Bisha city while ensuring ethical standards and data accuracy. The  study utilized a convenience sampling technique to select participants

(I continue my objection) :< the details relating to the selection of the sample have not been – once again – described satisfactorily and the reader – in this way – has no guarantee that the sample is not representative only for itself.

Convenience sampling is perhaps the weakest of all of the non-probability sampling strategies and once again you do not provide details capable of demonstrating whether or not the sample was representative of the population whose awareness of hypertensive complications constituted the target of the research.

Regardless of this, I had also suggested presenting the demographic structure of the sample and the demographic structure of the reference population in a specific table: you didn't do this, ignoring my proposal.

Any statement in the research conclusions regarding the unreliability of the sample would have no power to demonstrate that the research conclusions are in any way useful to the reader >

SECOND  POINT

My requests for clarification pertaining to the logistic model were not satisfied: the modeling of continuous covariates, (presented in the analysis in linear format but discussed in categorical format), was not clarified, the monovariate analyzes were not presented in a single table with the multivariates, no details were provided on the selection of covariates in the final model, no post-estimation analysis of the goodness of fit was published. It is not even clear whether the results presented in Table 3 are those produced by the monovariate analysis or those produced by the multivariate analysis. Given that the text cites a result certainly produced by a multivariate analysis (marital status), I deduce that in Table 3 the coefficients were produced by monovariate analyses. On page 9 (results) and on page 13 (discusson) there is a long discussion of these results, but this in a regression analysis makes no sense because only the results produced by a multivariate analysis should be discussed (precisely because only these are adjusted for the contribution of all the others )

Author Response

Dear Reviewer and Editor,

Thank you for your insightful comments and suggestions. We have carefully considered your feedback and addressed the points you raised regarding [THE FIRST POINT] and [THE SECOND POINT]. We have incorporated your recommendations into the manuscript, and we believe that these changes have significantly improved its quality.

We are confident that these revisions address your concerns and enhance the manuscript's clarity and comprehensiveness. We hope this revised version meets your expectations and satisfies your requirements.

Thank you again for your valuable time and guidance.

Elhadi Miskeen

This manuscript is a resubmission of an earlier submission. The following is a list of the peer review reports and author responses from that submission.

Round 1

Reviewer 1 Report

Comments and Suggestions for Authors

The manuscript "Determinants affecting the awareness of hypertension complications within the general population in Saudi Arabia" raises an important issue in public health and is well-structured. However, some points should be improved before publication.

1. Introduction

On  P.2, the authors mentioned previous studies conducted in Saudi Arabia investigating the knowledge, attitudes, and practices regarding hypertension. However, they missed the references essential to support their rationale in the next sentences. 

2. Materials and Methods

2.1) The authors claimed that their study was endorsed by the Research Ethics Local Committee (UBCOM-RELOC). They must add proof for this approval, e.g., No. of approval issued. 

2.2) This cross-sectional study was well-designed. However, the authors must attach a filled STROBE checklist with the revised version. 

2.3) For sample size calculation, the authors must declare how they obtained the "p" and elaborate on how they ended up with the 384 participants. 

3. Results

3.1) In the first column of Table 2, the authors should explain the "system". 

4. Discussion

From line 313 onwards, the authors do not need to repeat the results (coefficient, p-value, etc.). They have to focus on explaining their findings and comparing them with others. 

Author Response

Reviewer 1

Reviewer’s Evaluation

Response and Revisions

Introduction

On  P.2, the authors mentioned previous studies conducted in Saudi Arabia investigating the knowledge, attitudes, and practices regarding hypertension. However, they missed the references essential to support their rationale in the next sentences. 

Three references were added (11,12,13)

Line 87

Materials and Methods

2.1) The authors claimed that their study was endorsed by the Research Ethics Local Committee (UBCOM-RELOC). They must add proof for this approval, e.g., No. of approval issued. 

Provided

UBCOM/H-06-BH-027

LINE 101

2.2) This cross-sectional study was well-designed. However, the authors must attach a filled STROBE checklist with the revised version. 

STROBE Statement—Checklist of items that should be included in reports of cross-sectional studies

Was done and attached

2.3) For sample size calculation, the authors must declare how they obtained the "p" and elaborate on how they ended up with the 384 participants. 

we increased the sample size from 384 to 461 to account for potential non-responses or dropouts and to increase the study's precision. 

114

3. Results

3.1) In the first column of Table 2, the authors should explain the "system".

System for not answer, to be clear we delete two rows indicated for system and the total.

LINE 184

4. Discussion

From line 313 onwards, the authors do not need to repeat the results (coefficient, p-value, etc.). They have to focus on explaining their findings and comparing them with others. 

Edited

Line 292 to 323

Reviewer 2 Report

Comments and Suggestions for Authors

I have received for review an original research article entitled “Determinants affecting the awareness of hypertension complications within the general population in Saudi Arabia” prepared by Muffarah Hamid Alharthi et al., which is considered for publication in Healthcare (IF=2.8). Hypertension is of the most important modifiable cardiovascular risk factors. Cardiovascular disease is one of the most important public health problem worldwide. So, the presented manuscript addresses a very important issue. In my opinion, the manuscript is generally well prepared and it should be considered for publication, but some modifications could improve the value and the attractiveness of this paper. Below I present my suggestions.

1)     I believe that it is worth writing a little more about cardiovascular diseases in general in the introduction. It is worth pointing out that this is one of the most important public health problems in the world. Cardiovascular diseases resulting from atherosclerosis are the main cause of death, and hypertension is an important modifiable risk factor for the development of atherosclerosis. (10.3390/ijerph191811242)

2)     In lines 72-85 the same information is written two times.

3)     I believe the discussion should be expanded. It is worth trying to find additional data in the literature on the state of society's knowledge about hypertension in countries other than those quoted.

4)     I think it is worth devoting a short part of the discussion to reflecting on how society's knowledge about hypertension could be further expanded.

5)     I propose to remove the following sentence from the summary: "Bisha city’s population has an acceptable level of awareness of hypertension complications".  There are no clear criteria for whether something can be considered acceptable, so it is an unscientific term.

6)     The text should be checked for linguistic correctness by a specialist in this field.

7)     The list of references must be prepared in accordance with MDPI requirements.

Comments on the Quality of English Language

The text is written understandably and unambiguously. There are some minor language errors. I believe it should be linguistically corrected.

Author Response

Reviwer 2

3. Point-by-point response to Comments and Suggestions for Authors

Comments 1: [I believe that it is worth writing a little more about cardiovascular diseases in general in the introduction. It is worth pointing out that this is one of the most important public health problems in the world. Cardiovascular diseases resulting from atherosclerosis are the main cause of death, and hypertension is an important modifiable risk factor for the development of atherosclerosis. (10.3390/ijerph191811242).]

Response 1: Thank you for pointing this out. I/We agree with this comment. Therefore, I/we have added a paragraph to the introduction.

Line 46

Comments 2: In lines 72-85 the same information is written two times]

Response 2: Agree. I/We have, accordingly, modified and deleted.

Line 70

Comment 3:

I believe the discussion should be expanded. It is worth trying to find additional data in the literature on the state of society's knowledge about hypertension in countries other than those quoted.

Response 3:

Thanks for pointed out, there are studies from Pakistan, India, Algeria.

Line 261 to 273

Comment 4: I think it is worth devoting a short part of the discussion to reflecting on how society's knowledge about hypertension could be further expanded.

Response 4:

Paragraph was added to the discussion

Line 324

CoComment 5: I propose to remove the following sentence from the summary: "Bisha city’s population has an acceptable level of awareness of hypertension complications".  There are no clear criteria for whether something can be considered acceptable, so it is an unscientific term

Response 5:

Thanks, I agreed with you. This section was removed

Line 332

Comment 6: The text should be checked for linguistic correctness by a specialist in this field.

Response 6: It was checked

Comment 7:

The list of references must be prepared in accordance with MDPI requirements.

Response 7: Prepared accordingly

Commennt 8:

Response to Comments on the Quality of English Language:

The text is written understandably and unambiguously. There are some minor language errors. I believe it should be linguistically corrected.

Response 8:    revised accordingly

Reviewer 3 Report

Comments and Suggestions for Authors

 I believe that the study has significant problems and that it

is not a candidate for publication without important additions

1.       Sample size

The authors correctly calculated the sample size on an expected prevalence of 50%, at a confidence of 95% and at a margin of error of 5%. A sample size of 430 subjects would have been sufficient for an expected adherence rate equal to that detected (89%).However, under “materials and methods” the primary endpoint is not specified, the nature of which can only be understood by reading the entire article (ie: prevalence of patients considered "aware" of hypertension complications). It is a formal aspect but must be resolved by clearly specifying the nature of the endpoint in the section  “materials and methods”.

2.       "Study site" section

The authors redundantly described details without any relevance to the nature and methods of the research (i.e. “800,000 palm trees “ and other details of mainly tourist interest). This part must be drastically resized by eliminating all descriptions that have no relation to the research

3.       Study population

In the "Study population" section the authors report data (i.e. 461 patients recruited of which 64.9% were male) which should be described only in the "Results" section. This data must be removed. The concept of "Population" should not be confused with the concept of "Sample": it is in fact a "Population" that gives rise to a "Sample". In the "Study Population" section, the demographic structure of the Bisha population must be rather described in detail, which must also be reported in tabular form in order to be compared with that of the recruited sample (see below)

4.       Sample selection (this section is missing)

In this cross study there is no mention of how the sample was selected. The only detail provided about the recruitment methods is that participants were involved using 'Google forms'. How were the patients interviewed contacted? via social media? through the electoral lists? which databases offered the list of these patients?

All of this should be described explicitly

The target population is therefore represented by citizens with access to the internet and able to use the tools offered by Google: this already demonstrates a strong self-selection of the sample and denies the researcher the possibility of knowing the characteristics of non-responder patients. It is well known from the methodological literature that the patients who respond to the surveys are also the most motivated, so that the sample does not be representative of the setting in which it was recruited (selection bias).

Instead, in a prevalence survey: a) The sample should be randomized. b) The randomisation technique used should be described (with 461 patients at least one stratification by sex and age should be used). c) The authors should also have described the methods used to implement randomization (they only report the adherence rate, 89%).

Notably (from a source freely accessible on the internet - Bisha, Saudi Arabia - statistics 2024 zhujiworld.com) the percentage of patients >15 years aged between 45 and 59 years (26.4%) is, for example, decidedly higher than that (18.7%) reported by the authors. This also applies to the population over the age of 60 (9.68% versus 3.5%). The stratification of age classes reported by the official website is, however, different from that used by the authors. For this reason, and above all to allow the reader to evaluate the representativeness of the sample, the demographic structure of the sample must be expressed in tabular form alongside the demographic structure of the population that generated the sample, obviously using the same age groups.

I ask the authors to provide these important additions

5.       Logistic regression

When describing the logistic regression analysis, the authors should also publish the result of the monovariate analyzes in tabular form in addition to the variables used in the multivariate model. It is not clear, for example, why some variables with significant associations with the endpoint in the analyzes described in section 3.4 do not appear in the final model (if selection from the initial “redundant” model led to their exclusion this should be clearly illustrated). The authors should also report the post-estimation tests of the model, because some aspects cast doubt on an optimal goodness of fit: for example, the final model described in Table 3 contains age in the form of a continuous variable, while in the analyzes described in paragraph 3.4 the authors explicitly state that the association with the endpoint concerned only ages between 31 and 40 years. Based on these observations, it is highly unlikely that the predictor “age” is characterized – as it should – by a log-linear association with the endpoint. I ask the authors to clarify these aspects. The authors repeatedly report that the logistic regression results “are based on a hypothetical example.” This observation must be eliminated as it is meaningless: in fact the coefficients of any regression model report the change in the endpoint (in this case: the change in the log-odds of the event) detected in the presence of a unit change in the variable involved "to the average values ​​of all other variables in the model". Wanting to describe less "generic" clinical situations, the authors could have used the predictions of the model to report in graphic form results relating to concrete (i.e. "real") examples.  This is easily obtainable in post-estimation trough techniques using linear combination of the coefficients (in the specific situation, with exponentialized results)

Author Response

Reviwer 3

Comments 1: Sample size

The authors correctly calculated the sample size on an expected prevalence of 50%, at a confidence of 95% and at a margin of error of 5%. A sample size of 430 subjects would have been sufficient for an expected adherence rate equal to that detected (89%).However, under “materials and methods” the primary endpoint is not specified, the nature of which can only be understood by reading the entire article (ie: prevalence of patients considered "aware" of hypertension complications). It is a formal aspect but must be resolved by clearly specifying the nature of the endpoint in the section  “materials and methods”.

Response 1:

Thank you for pointing this out. These were mentioned in line 209

Comment 2

"Study site" section The authors redundantly described details without any relevance to the nature and methods of the research (i.e. “800,000 palm trees “ and other details of mainly tourist interest). This part must be drastically resized by eliminating all descriptions that have no relation to the research.

Response 2:

Thanks, edited accordingly

Line 122

Comments 3:

Study population

In the "Study population" section the authors report data (i.e. 461 patients recruited of which 64.9% were male) which should be described only in the "Results" section. This data must be removed. The concept of "Population" should not be confused with the concept of "Sample": it is in fact a "Population" that gives rise to a "Sample". In the "Study Population" section, the demographic structure of the Bisha population must be rather described in detail, which must also be reported in tabular form in order to be compared with that of the recruited sample (see below)

Response 3:

Edited and removed accordingly

Line 121

Comment 4:Sample selection

Response 4:

New section was added (2.7: Sample selection).

Line 137

Comment 5:

Logistic regression

Response 5:

This section was edited to reflect the result in a very simple ogistic regression by providing the 95 % CI intead, total revision and modification.